# Aging of Thin-Film Composite Membranes Based on Crosslinked PTMSP/PEI Loaded with Highly Porous Carbon Nanoparticles of Infrared Pyrolyzed Polyacrylonitrile

**DOI:** 10.3390/membranes10120419

**Published:** 2020-12-14

**Authors:** Danila Bakhtin, Stepan Bazhenov, Victoria Polevaya, Evgenia Grushevenko, Sergey Makaev, Galina Karpacheva, Vladimir Volkov, Alexey Volkov

**Affiliations:** A.V. Topchiev Institute of Petrochemical Synthesis RAS, Russian Academy of Sciences, Moscow 119991, Russia; db2@ips.ac.ru (D.B.); sbazhenov@ips.ac.ru (S.B.); polevaya@ips.ac.ru (V.P.); evgrushevenko@ips.ac.ru (E.G.); makaev@ips.ac.ru (S.M.); gpk@ips.ac.ru (G.K.)

**Keywords:** crosslinked PTMSP, thin-film composite membranes, aging, infrared pyrolyzed polyacrylonitrile, porous carbon filler, hybrid materials

## Abstract

The mitigation of the physical aging of thin-film composite (TFC) poly[1-trimethylsilyl-1-propyne] (PTMSP) membranes was studied via the simultaneous application of a polymer-selective layer crosslinking and mixed-matrix membrane approach. For the first time, a recently developed highly porous activated carbon material (infrared (IR) pyrolyzed poly[acrylonitrile] (PAN) or IR-PAN-a) was investigated as an additive to a PTMSP-selective layer for the reduction of aging in TFC membranes. The total electric energy spent on the IR irradiation treatment of IR-PAN-a particles was twice lower than conventional heating. The flat-sheet porous microfiltration membrane MFFK-1 was used as a support, and the crosslinked PTMSP/PEI loaded with a porous filler was applied as a selective layer (0.8–1.8 µm thick) to the TFC membranes. The initial IR-PAN-a sample was additionally milled to obtain a milled IR-PAN-aM sample with a monomodal particle size distribution of 500–800 nm. It was shown that IR-PAN-a, as a filler material with a high surface area and pore volume (2450 m^2^/g and 1.06 cm^3^/g, respectively) and a well-developed sponge-like structure, leads to the increase of the N_2_, O_2_, and CO_2_ permeance of PTMSP-based hybrid membrane material and the decrease of the aging of PTMSP. The simultaneous effect of crosslinking and the addition of a highly porous filler essentially improved the aging behavior of PTMSP-based TFC membranes. The monomodal and narrow particle size distribution of highly porous activated IR-pyrolyzed PAN is a key factor for the production of TFC membranes with reduced aging. The highest stability was achieved by the addition of a milled IR-PAN-aM sample (10 wt%). TFC membrane permeance was 6300 GPU (30% of initial permeance) after 11,000 h of aging at ambient laboratory conditions.

## 1. Introduction

Most polymeric membranes used for gas separation are based on glassy polymers due to their good balance between selectivity and permeability as well as their ability to withstand processing conditions [1,2,3,4,5]. However, glassy polymers possess an unrelaxed fractional free volume (FFV) due to their rigid structure and undergo physical aging because of macrochain rearrangements, resulting in the deterioration of membrane permeability with time (the so-called “aging effect”). Therefore, the effect of physical aging is significantly more pronounced for high-free-volume glassy polymers and is actively studied in the literature [6,7,8,9,10,11,12,13,14,15,16,17,18,19,20,21,22,23,24,25,26,27,28,29,30].

The stabilization of the transport properties of highly permeable glassy polymers can be improved by the restriction of the mobility of macrochains. Three main pathways are usually explored: chemical crosslinking [10,12,19,24], the introduction of various additives [11,13,14,15,16,17,22,25,26,28,31], and a combination of these two approaches [8,9]. The crosslinking of a polymeric matrix is usually associated with a certain decrease in the fractional free volume and gas permeability. However, the simultaneous introduction of fillers might lead to the appearance of defects on the interface between polymer and filler particles, resulting in an increase in gas permeability. For instance, it was shown that the introduction of 35 wt% fumed silica nanoparticles or TiO_2_ in poly(trimethylsilylpropyne) (PTMSP) or poly(methylpentyne) (PMP), together with crosslinking, yielded a stable performance and gas permeances higher than those of neatly corresponding polymers [8,9].

Porous fillers of a hydrophobic nature with nanometer-scale pores and a uniform distribution in the polymeric matrix can lead to a certain restriction of the mobility of macrochains due to their interaction with polymer segments and, on the other hand, boost the FFV and gas permeance through the membrane, providing additional pathways for gas transport through such fillers. Porous aromatic frameworks (PAFs) with different structures and functional groups have gained a great deal of attention in recent years from this perspective [13,14,15,16,17,26,28,29,30]. The introduction of PAF particles with a size of up to a few hundred nanometers in high-free-volume glassy polymers increases gas transport and noticeably mitigates the physical aging of the membrane material because of the increase of the FFV and the partial intrusion of macrochain segments into the pores of the nanoparticles. Such an approach was successfully demonstrated by the introduction of fillers of the PAF-1 family into dense PTMSP-, PIM-1-, and PMP-based films with a thickness of 100 μm [13,14,15,16,26] and PAF-11 and PTMSP films with a thickness of up to 40 μm [17].

It should be noted that selective layers based on PIMS-like materials in gas-separation membranes must be at the micron or submicron level to ensure a high flux performance. However, glassy polymers such as phenylene oxide, polysulfone, polyimide, and polyacetylenes in the form of thin films, with a thickness of up to a few microns, undergo accelerated physical aging [32,33,34]. Therefore, it is important to study the stabilization effect of different fillers on the transport properties of selective layers in thin-film composite (TFC) membranes, as was recently reported for a PTMSP/PAF-11 system [28], which demonstrated the maximum achievable membrane performance stability over time as follows: CO_2_ permeance of 1900 GPU and ideal O_2_/N_2_ and CO_2_/N_2_ selectivities of 1.7 and 6.7, respectively.

A new synthesis method for the preparation of highly porous polymer-derived microporous carbon materials using infrared (IR) irradiation has recently been developed [35,36]. A nitrogen-containing highly porous carbon material, named IR-PAN-a, synthesized by the IR heating of polyacrylonitrile (PAN) with the activation of polymer transformation by KOH, was reported for the first time by the authors of [35]. It is important to emphasize that IR synthesis provides a considerable reduction of activation time (2 min at 800 °C). IR irradiation treatment is also an energy- and time-efficient synthesis method for the fabrication of thermally rearranged (TR) and carbon molecular sieve (CMS) membranes for gas separation [37,38,39].

According to N_2_ adsorption measurements, the apparent BET surface area and pore volume of PAF-11 are equal to 240 m^2^/g and 0.35 cm^3^/g, respectively [29]. At the same time, the apparent BET surface area and pore volume of IR-PAN-a are equal to 2450 m^2^/g and 1.06 cm^3^/g, respectively [35]. Therefore, the pore volume and apparent surface area of IR-PAN-a are, respectively, 3 and 10 times higher than those of PAF-11.

The authors of [12,14,22] demonstrated that the PTMSP side-chain can partially penetrate the pores of PAF-1, resulting in the rigidity of the overall polymer structure and preventing physical aging in the membranes. This process was recently termed porosity-induced side-chain adsorption (PISA) [40]. One can assume that applying additives with a significantly larger pore volume and surface area can improve the effect of side-chain intrusion into the additive’s porous structure, resulting in stable membrane performance with time. Furthermore, as mentioned by the authors of [22], PAF-like materials can be considered promising additives toward the reduction of physical aging in polymers of intrinsic microporosity, but the synthesis of such materials is quite complex and hinders commercial implementation. Therefore, investigations were conducted on the addition of hypercrosslinked polymers into PTMSP, i.e., poly(dichloroxylene) (p-DCX) [22,40] and hypercrosslinked polystyrene (HCL-PS) [41], which were first synthesized by Davankov et al. [11,25,26,27,28,29,30,31,32,33,34,35,36,37,38,39,40,41,42]. From this perspective, a new type of IR-derived porous additive (IR-PAN) could be a promising alternative due to its energy- and time-efficient synthesis method. For example, the applicability and advantages of efficient IR irradiation for preparing thermally rearranged and carbon molecular sieve membranes for gas separation were confirmed by the authors of [37]. IR heating has considerable advantages over traditional conventional heating by providing energy saving by a factor of 10 and an increase of the overall productivity of the same membranes by a factor of 100. The authors of [43] also reported that the replacement of the conventional heating of PAN membranes with IR heating reduced the treatment time from 6 h at 250 °C to 5 min at 170 °C (estimated energy saving by a factor of 6.5).

With this in mind, in this work, IR-pyrolyzed PAN was investigated for the first time as a new type of filler to prevent physical aging in both dense and TFC PTMSP membranes. Gas transport properties were monitored during 500 h of annealing at 100 °C (dense membranes) and long-term monitoring at ambient conditions (TFC membranes).

## 2. Materials and Methods

### 2.1. Chemicals

Polyethyleneimine (PEI) (Mw = 25,000, Mn = 10,000), poly(ethylene glycol) diglycidyl ether (PEGDGE) (Mn = 500), methanol (≥99.6%) (Sigma-Aldrich, St. Louis, MO, USA), chloroform, and 1-butanol (chemical grade, Chimmed, Russia) were taken without further purification.

### 2.2. Synthesis of PTMSP

PTMSP (Mw = 900,000, Mw/Mn = 1.9, [η]25toluene = 6.0 dL/g) was synthesized in a toluene solution at 25 °C using TaCl_5_ and triisobutylaluminum (TIBA) as a catalyst and cocatalyst, respectively, according to the procedure described in [44].

### 2.3. IR-pyrolyzed PAN Materials

A new synthesis method for the preparation of highly porous polymer-derived microporous carbon materials using IR irradiation was recently developed [35,36]. In accordance with the protocol described in [35], PAN was first treated at 200 °C under IR heating in the air for 20 min. The obtained powders were impregnated with a KOH aqueous solution (ratio of precarbonized polymer/KOH = 1:1 *w*/*w*). The suspension was exposed for 24 h and dried at 80 °C in a vacuum until a constant weight, followed by a IR heating stage at 800 °C for 2 min in a nitrogen atmosphere. PAN (Mn = 79,000, Mw/Mn = 3.54) was synthesized in an aqueous medium in the presence of ammonium persulfate and sodium dithionite at 60 °C.

In this work, the IR-PAN-a sample was additionally ground in an HDDM/01 bead mill (Union Process, Akron, OH, USA) for 2 h at a stirrer speed of 600 rpm (bead material: ZrO_2_, 5 mm) (IR-PAN-aM). The IR-PAN-a sample was suspended before grinding in chloroform with a concentration of 0.1 wt%.

### 2.4. Membrane Fabrication

#### 2.4.1. Dense Membranes

The PTMSP/IR-PAN casting solutions with a filler content of 10 wt% were prepared by mixing 0.5 wt% PTMSP and IR-PAN solutions in chloroform. IR-PAN solutions were placed in an ultrasonic bath for 15 min before mixing with polymer solutions. The resulting PTMSP/IR-PAN solutions were stirred with a magnetic stirrer for 15 min and placed in the ultrasonic bath for another 15 min before membrane casting. The PTMSP/IR-PAN solutions were cast onto cellophane support and blanketed by a glass plate for the slow evaporation of solvents at ambient conditions (about 200 h) until a constant weight of the samples was attained. The resulting thicknesses of the PTMSP/IR-PAN dense membranes were in the range of 35–40 µm. The membrane thicknesses were measured by a Mitutoyo^®^ electronic micrometer.

#### 2.4.2. Thin-Film Composite Membranes

The flat-sheet porous microfiltration MFFK-1 membrane (ZAO STC “Vladipor”, Russia) was used as a support for the fabrication of TFC membranes. The microfiltration layer of the MFFK-1 membrane is composed of a tetrafluoroethylene/vinylidenefluoride copolymer (PTFE/PVDF fluoroplastic F-42L) on a polypropylene nonwoven support. Prior to the preparation of the TFC membranes, all supports were washed in ethanol for 24 h and dried in a Binder airflow oven at 60 °C until a constant weight was attained.

Figure 1 shows the scheme of the fabrication of TFC membranes. The membranes were prepared by the kiss-coating (or meniscus-coating) technique, which is a modified version of the dip-coating procedure. This approach provides one-sided contact between the surface of the porous support and the casting solution. In accordance with the procedure described elsewhere [45], coating was performed at room temperature (23 ± 2 °C) using the lab-scale setup, which involves the casting stage with the support transport mechanism (size of the initial support sheet was 0.1 × 1 m). Prior to coating, the support was impregnated with ethanol and then water to prevent the penetration of the polymeric layer into the pores of the MFFK-1 support. The desired speed of the porous support movement was fixed by the speed controller (0.1−0.7 m/min), and the polymer solution was deposited by coating. The casting PTMSP solutions containing 4 wt% PEI (relative to the weight of PTMSP) were prepared by mixing 0.5 wt% solutions of pure polymers in chloroform. In the case of the loaded PTMSP/PEI layer, IR-PAN additives were added to the polymeric casting solution. PTMSP/PEI/IR-PAN casting solutions with different filler contents (10, 20, and 30 wt%) were prepared by mixing 0.5 wt% PTMSP/PEI and IR-PAN solutions in chloroform. The IR-PAN solution was placed in an ultrasonic bath for 15 min before mixing with polymer solutions. The resulting PTMSP/PEI/IR-PAN solutions were stirred with a magnetic stirrer for 15 min and placed in an ultrasonic bath for another 15 min before membrane casting. After the deposition of a thin selective layer on the MFFK-1 support, PEI crosslinking was conducted by immersing the prepared composite membrane for 72 h in a solution of PEGDGE crosslinking agent (4 wt%) in methanol. After that, the membranes were first held in pure 1-butanol for 24 h and then in a 25 wt% solution of ethanol in water for at least 24 h and dried. All TFC membranes prepared in this work are presented in Table 1.

### 2.5. Gas Permeability Measurement and Aging

#### 2.5.1. Dense Membranes

Single gas permeation measurements (N_2_, O_2_, and CO_2_) were carried out at a temperature of 30.0 ± 0.1 °C and a feed pressure of 0.05–0.8 bar, using a constant volume/variable pressure experimental setup (Time-Lag Machine, Helmholtz-Zentrum, Geesthacht, Germany) [46]. The measurements were performed for the as-cast PTMSP and PTMSP/IR-PAN-1 membranes and the same membranes after their annealing in air at 100 °C for 200 and 500 h. The permeability coefficient *P* expressed in Barrer (1 Barrer = 7.5 × 10^−18^ m^3^·m·m^−2^·s^−1^·Pa^−1^) was estimated by the linear extrapolation of experimental data to zero transmembrane pressure. The ideal selectivity of the pair of gases was calculated as the ratio of the permeabilities of individual gases.

#### 2.5.2. TFC Membranes

The transport characteristics of TFC membranes in time (aging) were monitored by the individual gas permeability (N_2_, O_2_, CO_2_) measured by a volumetric method at room temperature using the setup described in [30]. The feed and permeate pressure was up to 2 bar and 1 bar, respectively. The active membrane surface area S was 12.6 cm^2^. In all cases, the gas flux was increased linearly with respect to transmembrane pressure, which was allowed to determine the gas permeance P/l, expressed in GPU (1 GPU = 10^−6^ cm^3^(STP)·cm^−2^·s^−1^·cmHg^−1^), as a graph slope. The TFC membranes aging at room temperature were monitored for up to 11,000 h. The aging time of 11,000 h was chosen for convenience of comparison of our results with previously reported data on aging of TFC membranes based on PTMSP loaded with PAF-11 (aging time: 10,800 h) [28]. All gas permeances presented in this work were recalculated according to STP conditions. The relative gas permeance *P*/*l′* and relative ideal gas selectivity *α′* for aged membranes were calculated as follows:(1)$$P^{\prime }=\frac{P_{aged}}{P_{as\_cast}},$$
(2)$$\alpha^{\prime }=\frac{\alpha_{aged}}{\alpha_{as\_cast}},$$
where *P_as_cast_* and *α_as_cast_* corresponded to gas permeance and ideal selectivity for the as-cast membranes, and *P_aged_* and *α_aged_* correspond to aged membranes, respectively.

### 2.6. Scanning Electron Microscopy (SEM)

The morphology of the synthesized composite membranes was studied by high-resolution SEM on a TM3030Plus tabletop microscope (Hitachi, Tokyo, Japan). To prepare membrane cleavages, the membranes were preliminarily impregnated in isopropanol and then broken in a liquid nitrogen medium. Using a DSR-1 table spraying gun (Nanostructured Coatings Co., Tehran, Iran), the prepared samples were coated with a thin (5 nm thick) layer of gold in a special vacuum chamber (about 50 torr).

### 2.7. Particle Size Distribution

The size distributions of the IR-PAN particles were determined by dynamic light scattering on a Malvern Zetasizer Nano analyzer (Malvern Panalytical Ltd., Malvern, UK). The test samples were prepared by suspending the particles in n-hexane (0.06 g of a sample in 10 mL of solvent).

## 3. Results and Discussion

### 3.1. Dense Hybrid Membranes Aging

IR-PAN-a is a highly porous carbon material with a sponge-like structure (Figure 2), inner spherical cavities of 0.5–2 μm with meso- and micropores distributed on their walls, an average micropore size of 0.66 μm, and an apparent BET surface area and pore volume of 2450 m^2^/g and 1.06 cm^3^/g, respectively [35]). Moreover, the total electric energy spent on the IR heating of IR-PAN-a particles was 2.95 kWh, whereas a conventional oven requires 5.72 kWh [47]. Assessment of the new additive’s potential for initiating the physical aging improvement of PTMSP-based hybrid materials was necessary at the first stage of the study. Dense PTMSP membranes containing 10 wt% IR-PAN-a were prepared for this purpose. We recently demonstrated the gas transport characteristics of a dense PTMSP membrane (films; thickness: 30–40 µm) containing 10 wt% PAF-11, which became stable upon annealing at 100 °C within a short time interval (100–200 h) [17]. However, for two other samples (1 and 5 wt%), gas permeability gradually decreased with time. In this study, the addition of 10 wt% of IR-PAN-a was chosen to conduct an accurate comparison of the effectiveness of the new additive.

The surface and transverse cleavage of these membranes are shown in Figure 3a,b. Dense PTMSP membranes without the addition of IR-PAN-a were also made. SEM images of the pristine dense PTMSP membrane are also presented in Figure 3. Traces of fine dust can be seen on the surface of the pristine dense PTMSP membrane (Figure 3c), and large agglomerates of particles up to several microns in size are visible on the surface and in the volume of the PTMSP membrane containing 10 wt% IR-PAN-a (Figure 3a,b). Table 2 represents the comparison of the gas transport characteristics between the original membranes, annealed for 200 and 500 h at 100 °C in an air atmosphere.

The data demonstrate that the addition of the highly microporous carbon material IR-PAN-a into PTMSP leads to a significant increase in the gas permeability coefficients for all the studied gases. For example, the addition of IR-PAN-a increases the CO_2_ permeability coefficient from 30,600 to 40,000 Barrer. It is also seen that an increase in the permeability coefficients for a hybrid membrane is accompanied by a slight decrease in the ideal selectivity. As expected, the annealing of membranes leads to a decrease in their gas permeability. However, two important advantages of the PTMSP + 10 wt% IR-PAN-a hybrid membrane should be emphasized.

First, the PTMSP sample lost its mechanical properties and collapsed after 500 h of annealing in air at 100 °C. This may indicate the occurrence of relaxation processes, as a result of which the number of auxiliary chains providing the mechanical strength of the polymer significantly decreased. At the same time, the hybrid PTMSP + 10 wt% IR-PAN-a membrane retained sufficient mechanical characteristics after 500 h of annealing in air at 100 °C, and the values of its gas permeability coefficients are presented in Table 2.

Second, for the hybrid membrane, a smaller relative decrease in gas permeation was observed after the first 200 h of annealing compared to the PTMSP membrane without IR-PAN-a additives. More importantly, the following 300 h of annealing practically did not change the transport properties of the hybrid dense PTMSP + 10 wt% IR-PAN-a membrane with a thickness of 38 µm.

It should also be noted that the ideal gas selectivity of the hybrid membrane sample increased during aging; as a result, the ideal selectivities became close to the values typical for a fresh film made of pure PTMSP. The increase in the selectivity of PTMSP membranes with aging is widely known [17,26].

It is interesting to compare the effectiveness of the new IR-PAN-a additive and the previously studied PAF-11 [17]. Figure 4 shows the data of the CO_2_ permeability coefficient for PTMSP hybrid dense membranes with IR-PAN-a and PAF-11 additives versus the annealing time at 100 °C.

As can be seen from Figure 4, the addition of IR-PAN-a leads to a higher CO_2_ permeability coefficient of the hybrid PTMSP/IR-PAN-a membrane compared to the PTMSP/PAF-11 membrane. In addition, the relative drop in CO_2_ permeability after 500 h of annealing at 100 °C for the sample with 10 wt% PAF was 30%, whereas the addition of IR-PAN-a reduced this drop to 20%. This effect is most likely associated with the increased specific surface area of IR-PAN-a (S_BET_ = 2450 m^2^/g [35]) compared to PAF-11 (S_BET_ = 240 m^2^/g [17]), as well as the sponge-like structure of IR-PAN-a with a network of macro-, meso-, and micropores [35]. That is why IR-PAN-a, as a filler material with a high specific surface area and developed sponge-like structure, leads to an increase in the permeability of the PTMSP-based hybrid membrane material (Table 2). Moreover, IR-PAN-a presumably acts, as it has been speculated for PAF fillers [17], as a “physical crosslinker” due to the adsorption and/or intrusion of polymer chain fragments into the porous structure of IR-PAN-a. This, in turn, leads to a significant decrease in the membrane aging effect. Therefore, it was of interest to study the introduction of this filler into the selective layer of the PTMSP composite membrane.

### 3.2. TFC Hybrid Membrane Aging

We investigated the effect of the number and size of IR-PAN-a particles on the gas transport properties of composite PTMSP membranes and their stability over time. Composite membranes were prepared on MFFK-1 microfiltration supports. In contrast to the traditionally used ultrafiltration supports, the MFFK-1 microfiltration support (average pore size: 250–410 nm) provides enhanced performance of the composite membrane [48]. Table 1 shows the names of the prepared membranes.

The original IR-PAN-a sample was processed in a bead mill (IR-PAN-aM) to study the effect of IR-PAN-a particle size on the transport properties of TFC membranes. Figure 5 shows the particle size distribution of IR-PAN-a and IR-PAN-aM, measured by the dynamic light scattering method. It can be seen that the original IR-PAN-a contains three groups of particles: small, medium, and large particles. The maximum diameters were 90, 650, and 6000 nm, respectively (Figure 5a). The particle size distribution of the milled sample IR-PAN-aM is monomodal with a rather narrow particle size distribution from 500 to 800 nm and, accordingly, a maximum particle diameter of about 650 nm (Figure 5b). The performed mechanical treatment of the sample made it possible to achieve the uniformity of the IR-PAN particle size; the influence of this factor was considered for the transport properties of PTMSP composite membranes.

Table 3 contains SEM micrographs of transverse cleavages of the investigated composite membranes. The sponge-like structure of the MFFK-1 porous support and the thin PTMSP-based layer on top of the porous support are clearly distinguishable for all membranes.

The change in CO_2_ gas permeance of the investigated composite membranes with time is shown in Figure 6. The permeance of the IR-PAN-a-loaded crosslinked membranes (for example, for CM-1–CM-6 P/l(CO_2_) = 21,000–25,000 GPU) significantly exceeds the permeance of the crosslinked composite PTMSP membrane (CM-7: P/l(CO_2_) = 15,000 GPU). The positive effect of IR-PAN as an additive can also be demonstrated by the reduction of the PTMSP aging; all IR-PAN-a-loaded samples of composite membranes (CM-1–CM-6) had higher gas permeance after 11,000 h of aging compared to a composite membrane with a selective layer of pure crosslinked PTMSP (CM-7). It is important to emphasize that the ideal gas selectivities of the PTMSP/IR-PAN composite membranes for CO_2_/N_2_ and O_2_/N_2_ were 3.5–4.0 and 1.3–1.5, lower than those of dense membranes for the same gas pairs: 5.2–5.8 and 1.4–1.5 (Table 2). This behavior is typical for the transport properties of TFC membranes and confirms the absence of defects in the thin selective layer of hybrid membranes. The results confirm that IR-PAN, as well as metal–organic frameworks and PAFs, stabilizes the gas transport properties of composite membranes with a thin selective PTMSP layer.

Taking into account the difference in the thickness of the composite membrane-selective layers (Table 3), the comparison of the relative gas permeance *P′* is a more accurate way to assess the effect of the filler on the aging of PTMSP (Table 4).

Reviewing Table 4, one can notice that the IR-PAN particle size and particle size distribution have a marked effect on the stability of the transport characteristics of TFC membranes. For the IR-PAN-a sample with a wide particle size distribution and the presence of three particle sizes, including large sizes (6000 nm) (Figure 4), there is no clear pattern between the amount of the additive and the stability of the membrane’s transport characteristics. The comparison of membranes CM-1 (10 wt% IR-PAN-a) and CM-7 (without an additive) indicates that the introduction of 10 wt% IR-PAN-a negatively affects the stability of the permeability of the composite membrane. The further introduction of 20 and 30 wt% IR-PAN-a leads to an increase and then a decrease in the stability of the permeability of the TFC membrane (membranes CM-2 and CM-3, respectively). There is no logic in the relative ideal selectivity value since this value should be higher for the CM-3 membrane and not for the CM-2 membrane. On the other hand, the use of the IR-PAN-aM sample with a narrow particle size range of 500–800 nm (Figure 4) leads to a steady improvement in the stability of membrane permeability in the entire studied range of additive concentrations (10, 20, and 30 wt%, membranes CM-4, CM-5, and CM-6, respectively). Moreover, the minimum studied additive content (10 wt%) ensured high gas transport characteristics and a minimal aging effect of the TFC membranes. A further increase in the additive content leads to a slight increase in the aging effect (albeit insignificant), as evidenced by a decrease in the P′ from 30.2 to 27.2%.

Therefore, monomodal and narrow particle size distribution of highly porous activated IR-pyrolyzed PAN is a key factor for the production of TFC membranes with reduced aging. In summary, 30% of the initial permeance with almost unchanged membrane selectivity (*α’* = 1.1) after 11,000 h of aging at ambient laboratory conditions were preserved by the introduction of IR-PAN-aM particles with a size of 500–800 nm into the selective layer from crosslinked PTMSP/PEI with a thickness of 1000 nm. For comparison, the composite membrane from PTMSP with the addition of 10 wt% PAF-11 on a PAN support preserved only 13% of the initial permeance after 11,000 h of aging [28].

## 4. Conclusions

For the first time, a recently developed highly porous activated carbon material was investigated as an additive to a PTMSP selective layer for the reduction of aging in TFC membranes. The flat-sheet porous microfiltration MFFK-1 membrane was used as a support, and the crosslinked PTMSP/PEI loaded with filler was applied as a selective layer of TFC membranes. IR-PAN-a, as a filler material with a high surface area and a developed sponge-like structure, was used for the preparation of dense membranes (thicknesses of 35–40 µm). An increase in the permeance of the PTMSP-based hybrid membrane material and a decrease in the aging of PTMSP were demonstrated.

The initial IR-PAN-a sample was additionally milled to obtain a milled IR-PAN-aM sample with a monomodal particle size distribution of 500–800 nm. It was shown that the minimum studied additive content (10 wt%) ensured high gas transport characteristics and a minimal aging effect of TFC membranes. A further increase in the additive content (20 and 30 wt%) leads to a slight increase in the aging effect (albeit insignificant), as evidenced by a decrease in the relative permeance P′ from 30.2 to 27.2%. Therefore, the monomodal and narrow particle size distribution of highly porous activated IR-pyrolyzed PAN is a key factor for the production of TFC membranes with reduced aging. In summary, 30% of the initial permeance after 11,000 h of aging in ambient laboratory conditions was preserved by the introduction of the IR-PAN-aM particles with a size of 500–800 nm into the selective layer from crosslinked PTMSP/PEI with thicknesses of 800–1000 nm. The resultant gas permeance values of this TFC membrane were 6300, 2000, and 1400 GPU for CO_2_, O_2_, and N_2_, respectively.

## Figures and Tables

**Figure 1 membranes-10-00419-f001:**
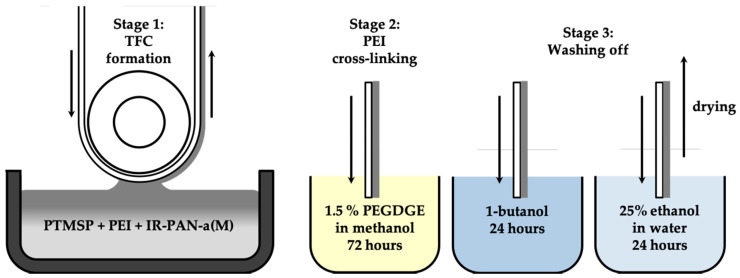
Scheme of thin-film composite (TFC) membrane fabrication.

**Figure 2 membranes-10-00419-f002:**
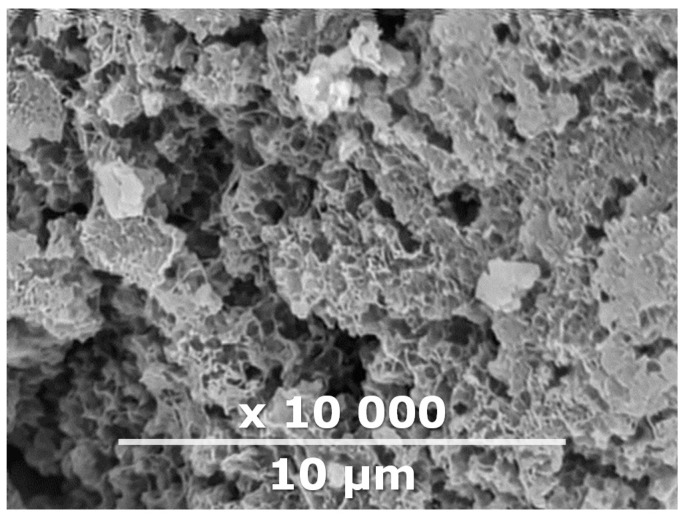
SEM images of IR-PAN-a.

**Figure 3 membranes-10-00419-f003:**
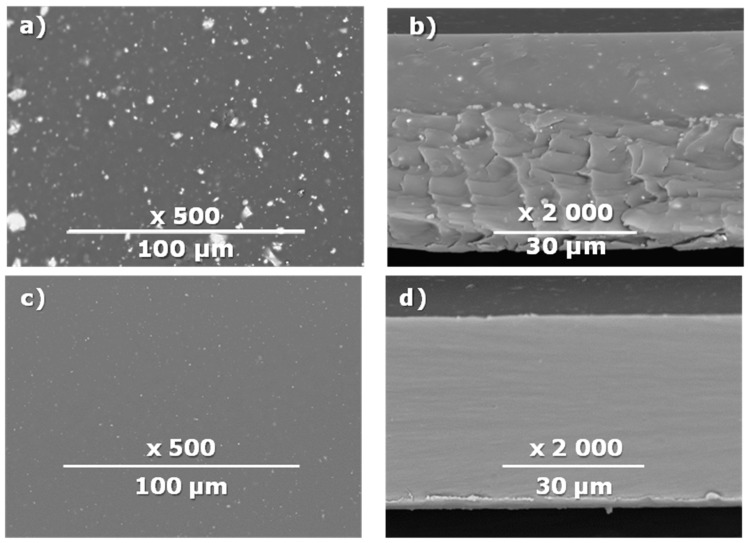
Surface (**a**,**c**) and transverse cleavage (**b**,**d**) of pristine poly[1-trimethylsilyl-1-propyne] (PTMSP) (**c**,**d**) and PTMSP films containing 10 wt% IR-PAN-a (**a**,**b**).

**Figure 4 membranes-10-00419-f004:**
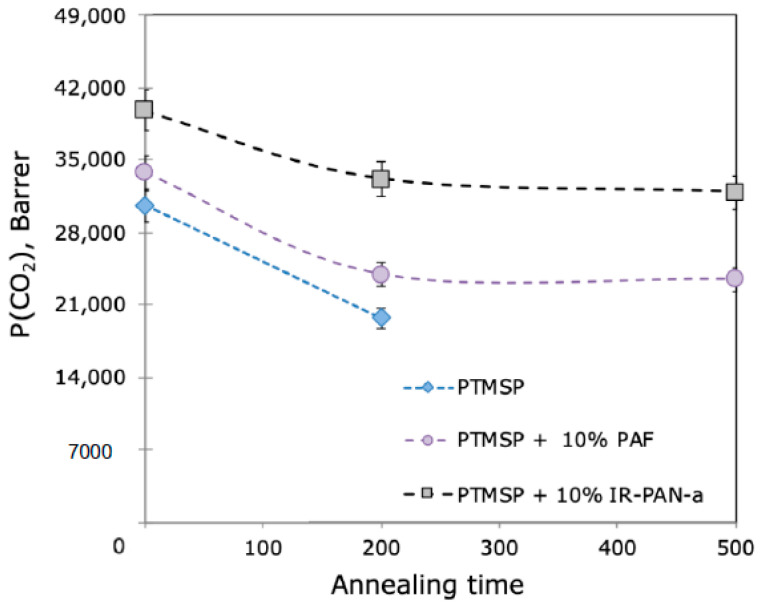
Impact of IR-PAN-a и PAF-11 [17] and annealing at 100 °C in air on the permeability coefficient of hybrid dense PTMSP membranes.

**Figure 5 membranes-10-00419-f005:**
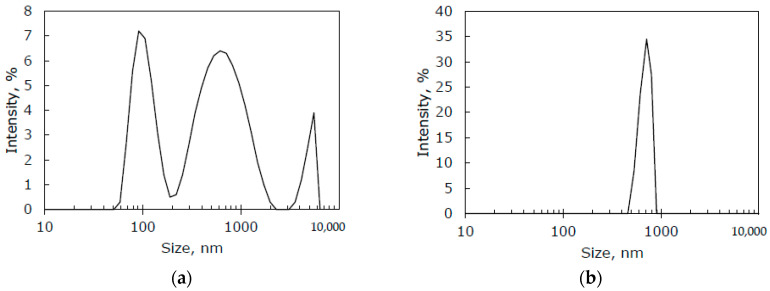
Particle size distribution of IR-PAN-a (**a**) and IR-PAN-aM (dynamic light scattering data) (**b**).

**Figure 6 membranes-10-00419-f006:**
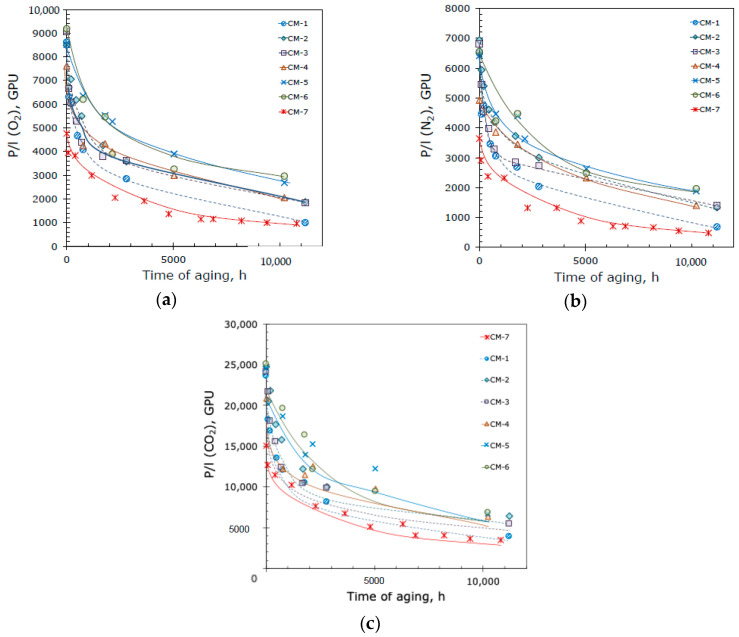
Gas permeance of TFC membranes without (CM-7) and with (CM-1–CM-6) IR-PAN filler: (**a**) O_2_, (**b**) N_2_, and (**c**) CO_2_.

**Table 1 membranes-10-00419-t001:** List of the TFC membranes.

Membrane Name	Crosslinking	Additive	Additive Content, wt%
CM-1	PEI + PEGDGE	IR-PAN-a	10
CM-2	PEI + PEGDGE	IR-PAN-a	20
CM-3	PEI + PEGDGE	IR-PAN-a	30
CM-4	PEI + PEGDGE	IR-PAN-aM	10
CM-5	PEI + PEGDGE	IR-PAN-aM	20
CM-6	PEI + PEGDGE	IR-PAN-aM	30
CM-7	PEI + PEGDGE	-	-

**Table 2 membranes-10-00419-t002:** Gas permeation performance of the PTMSP film.

Sample	Annealing Time at 100 °C, h	Thickness, µm	P, Barrer			α		Relative Change, %		
			N_2_	O_2_	CO_2_	CO_2_/N_2_	O_2_/N_2_	N_2_	O_2_	CO_2_
PTMSP	0	28.6	5500	8100	30,600	5.6	1.5	0	0	0
	200		3200	5000	19,800	6.1	1.6	41	38	35
	500		-	-	-	-	-	-	-	-
PTMSP + 10 wt% IR-PAN-a	0	38.0	7700	10,500	40,000	5.2	1.4	0	0	0
	200		5600	8600	33,200	5.9	1.5	27	18	17
	500		5500	8200	32,000	5.8	1.5	29	22	20

**Table 3 membranes-10-00419-t003:** SEM images and selective layer thicknesses for the thin-film composite PTMSP/IR-PAN membranes.

Composite Membrane Designation	Crosslinking Agents	Filler	Filler Content, % wt	SEM	Selective Layer Thickness, µm
CM-1	PEI+ PEGDGE	IR-PAN-a	10	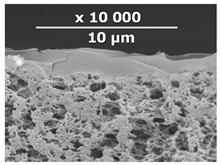	1.8
CM-2	PEI+ PEGDGE	IR-PAN-a	20	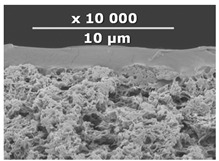	1.7
CM-3	PEI+ PEGDGE	IR-PAN-a	30	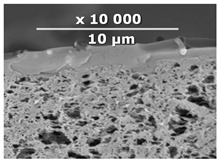	1.8
CM-4	PEI+ PEGDGE	IR-PAN-aM	10	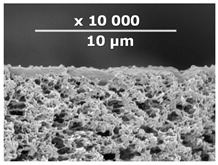	1.0
CM-5	PEI+ PEGDGE	IR-PAN-aM	20	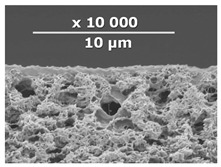	0.8
CM-6	PEI+ PEGDGE	IR-PAN-aM	30	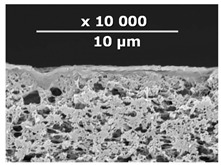	1.0
CM-7	PEI+ PEGDGE	-	-	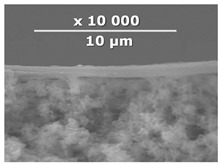	1.2

**Table 4 membranes-10-00419-t004:** Relative gas permeance *P′* and relative ideal selectivity *α′* for aged composite membrane.

Membrane	*P*′, %	*α*′
CM-1	16.6	1.6
CM-2	25.9	1.3
CM-3	22.4	1.1
CM-4	30.2	1.1
CM-5	27.2	0.9
CM-6	27.2	0.9
CM-7	23.2	1.6

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
