# Peer review of "Aging of Thin-Film Composite Membranes Based on Crosslinked PTMSP/PEI Loaded with Highly Porous Carbon Nanoparticles of Infrared Pyrolyzed Polyacrylonitrile"

_membranes, 2020, doi:10.3390/membranes10120419_

Round 1

Reviewer 1 Report

This is a very interesting proposal about thin-film composite (TFC) membranes based on PTMSP cross-linked with a recently developed highly porous activated carbon material (IR-PAN), in order to reduce aging in TFC membranes.

Thus, IR-pyrolyzed PAN was investigated as a new type of filler for the first time. TFC membranes with PTMSP top-layer loaded with IR-PAN were fabricated and the physical aging of TFC and dense membranes with virgin and IR-PAN-loaded PTMSP top-layer was evaluated at ambient conditions.

Despite the interesting topic and the novelty presented by this work, significant improvements must be made before being accepted for publication:

- Based on the introduction, it is understood that PTMSP-based TFC has accelerated physical aging and that, in this regard, the effect of PAF filler on the performance of TFC membranes has been previously reported. For example, PTMSP loaded with PAF 11 improving these properties, has been reported. The main difference between PAF-11 and the proposed new filler (IR-PAN) was described based on the significant increase in its surface area. Could it better support the hypothesis raised? How an increase in surface area or another property of IR-PAN is expected to improve the aging of modified membranes?. A better explanation and references for support should be included in the introduction.

- The modified membranes synthesis methodology must be substantially improved. Considering the various stages of modification to obtain the modified membrane, a detailed outline/scheme of the synthesis stages and/or an illustrative figure of the modified membrane identifying the formed/linked layers could help to better understand. The deposition of the thin layer on the MFFK-1 support and the cross-linking mechanism should be better explained and illustrated.

- In 2.5.2, must be TFC membrane instead of THC?. Moreover, it is describing that “the TFC membranes aging at room 160 temperature was monitored during up to 11,000 hours”. References on the selection of these times must be provided. 

- Line 185. Dense PTMSP membranes without the addition of IR-PAN should be shown, at least in supporting information.

- The behavior of the membranes modified with IR-PAN and PAF-11, both at 10% wt, was compared. Despite the significant differences in surface area between these materials (2450m2/g vs 240m2/ g, respectively), the relative drop in CO2 permeability was only 30% for PAF and 20% for IR-PAN. Could it then be considered that a high surface area does not guarantee a great improvement in the aging capacity as expected? A better comparison could be made based on the type of structures or other comparative property between these two materials for a better explanation.

- The particle size distribution of the ground IR-PAN-aM sample is monomodal with a fairly narrow particle size distribution of 500 to 800 nm obtained from the original IR-PAN-a containing three groups of particles: small particles, medium, and large (100, 800, and 8000 nm). Could you explain the phenomenon in grinding for the disappearance of the smallest particles, 100-500nm? In this procedure, it is ground and subsequently suspended in chloroform, what is the objective of this last step? These aspects could be included in the methodology and in the discussion.

- The discussion of results on the impact of the size and %wt of the fillers on the thickness of the composite membrane should be better presented. Membranes with ground IR-PAN-aM decreases the thickness with respect to modification with IR-PAN-a, however, it is not understood how it can be thinner than the pristine sample (CM-7). Moreover, how does the incorporation of 20%wt reduce the thickness with respect to 10 and 30% wt? What are the arguments for these trends? are these differences considered significant? These tendencies must be correlated with the performance of the membranes.

- There is no clear trend or argument for the impact of size and percentage filler on gas permeance and selectivity. In this sense, conclusions should include these relevant arguments of the study.

Author Response

Thank you very much for your time and effort to review our manuscript. We have made our best to improve our manuscript according to your comments.

Comment 1. Based on the introduction, it is understood that PTMSP-based TFC has accelerated physical aging and that, in this regard, the effect of PAF filler on the performance of TFC membranes has been previously reported. For example, PTMSP loaded with PAF 11 improving these properties, has been reported. The main difference between PAF-11 and the proposed new filler (IR-PAN) was described based on the significant increase in its surface area. Could it better support the hypothesis raised? How an increase in surface area or another property of IR-PAN is expected to improve the aging of modified membranes?. A better explanation and references for support should be included in the introduction.

Answer 1. The following has been added into introduction to support the hypothesis raised:

Authors of [14, 16, 24] demonstrated that the side-chain of the PTMSP can thread the pores of the PAF-1, providing rigidity to the overall polymer structure and preventing physical aging in the membranes. Recently, this process was termed as Porosity Induced Side chain Adsorption (PISA) [42]. One can assume that applying the additives with significantly larger pore volume and surface area can improve the effect of the side chain intrusion into additive’s porous structure, resulting in stable membrane performance in time. Besides, as it was mentioned in [24], whilst PAF-1 was an effective additive for reducing physical aging in mixed-matrix membranes, its complex synthesis route renders it too expensive for commercial uptake. Therefore, the investigations were carried out on the addition into PTMSP of hypercrosslinked polymers, i. e. poly(dichloroxylene) (p-DCX) [24, 42] and hypercrosslinked polystyrene (HCL-PS) [43], which were first synthesized by Davankov et al. [13, 27−45]. From this view point, a new type of IR derived porous additives (IR-PAN) might be a promising alternative due to a highly energy and time efficient synthesis method. For example, the applicability and advantages of efficient IR irradiation for preparing thermally rearranged and carbon molecular sieve membranes for gas separation were confirmed in [39]. By replacing the electric furnace with the IR furnace, energy savings by a factor of ~10 were obtained at the same thermal conversion rate and consequently, the overall productivity was improved by more than a factor of 100. It was also reported in [46] that replacement of conventional heating of PAN membranes by IR heating allowed to reduce the treatment time from 6 hours at 250°C down to 5 minutes at 170°C (estimated energy saving by a factor of 6.5).

Comment 2. The modified membranes synthesis methodology must be substantially improved. Considering the various stages of modification to obtain the modified membrane, a detailed outline/scheme of the synthesis stages and/or an illustrative figure of the modified membrane identifying the formed/linked layers could help to better understand. The deposition of the thin layer on the MFFK-1 support and the cross-linking mechanism should be better explained and illustrated.

Answer 2. The article was modified by addition of Figure 1, explaining TFC fabrication, and the following comment to this figure:

Figure 1 shows the scheme of TFC membranes fabrication. The membranes were prepared by the kiss-coating (or meniscus-coating) technique, which is a modified version of the dip-coating procedure. This approach provides a one-sided contact between the surface of the porous support and the casting solution. In accordance with the procedure described elsewhere [48], coating was performed at room temperature (23 ± 2 °C) using the lab-scale setup which involves the casting stage with the support transport mechanism (size of the initial support sheet is 0.1 x 1 m). Prior to coating, the support was impregnated with ethanol and then with water in order to prevent penetration of the polymeric layer into the pores of MFFK-1 support. The desired speed of porous support movement was fixed by the speed controller (0.1−0.7 m/min), and the polymer solution was deposited by coating. The casting PTMSP solutions containing 4 wt % PEI (relative to the weight of PTMSP) were prepared by mixing 0.5 wt % solutions of pure polymers in chloroform. In the case of loaded PTMSP/PEI layer, IR-PAN additives were added to polymeric casting solution. PTMSP/PEI/IR-PAN casting solutions with different filler content (10, 20 and 30 wt %) were prepared by mixing 0.5 wt % PTMSP/PEI and IR-PAN solutions in chloroform. IR-PAN solution was placed in the ultrasonic bath for 15 min before mixing with polymer solutions. The resulted PTMSP/PEI/IR-PAN solutions were stirred with magnetic stirrer for 15 min and placed in the ultrasonic bath for another 15 min before membrane casting. After the deposition of a thin selective layer on MFFK-1 support, PEI cross-linking was conducted via immersing the prepared composite membrane for 72 h into a solution of PEGDGE cross-linking agent (4 wt %) in methanol. After that, the membranes were first held in pure 1-butanol for 24 h, then in 25 wt % solution of ethanol in water for at least 24 h and dried. All TFC membranes prepared in this work are presented in Table 1.

Comment 3. In 2.5.2, must be TFC membrane instead of THC?.

Answer 3. Corrected in text

Comment 4. Moreover, it is describing that “the TFC membranes aging at room 160 temperature was monitored during up to 11,000 hours”. References on the selection of these times must be provided. 

Answer 4. In our previous work on TFC membranes [30. Bakhtin, D.S., Kulikov, L.A., Legkov, S.A., Khotimskiy, V.S., Levin, I.S., Borisov, I.L., Maksimov, A.L., Volkov, V.V., Karakhanov, E.A., Volkov, A.V. Aging of thin-film composite membranes based on PTMSP loaded with porous aromatic frameworks. J. Membr. Sci. 2018, 554, 211-220. https://doi.org/10.1016/j.memsci.2018.03.001.] PAN/PTMSP(6.8µm)+10%PAF-11 sample demonstrated the best results, leveling at the stable gas transport characteristics of 300, 500 and 1900 GPU for N2, O2 и CO2, respectively, after ~450 days (~ 10800 hours) of observation. In this work we chose the same period of time for convenience of comparison. The following explanation was added into the experimental part:

The aging time of 11000 hours was chosen for convenience of comparison of our results with previously reported data on aging of TFC membranes based on PTMSP loaded with PAF-11 (aging time 10800 hours) [30].

Comment 5. Line 185. Dense PTMSP membranes without the addition of IR-PAN should be shown, at least in supporting information.

Answer 5. The SEM images of pristine PTMSP dense membrane has been added in the figure 3.

Comment 6. The behavior of the membranes modified with IR-PAN and PAF-11, both at 10% wt, was compared. Despite the significant differences in surface area between these materials (2450m2/g vs 240m2/ g, respectively), the relative drop in CO2 permeability was only 30% for PAF and 20% for IR-PAN. Could it then be considered that a high surface area does not guarantee a great improvement in the aging capacity as expected? A better comparison could be made based on the type of structures or other comparative property between these two materials for a better explanation.

Answer 6. As it was mentioned above, simplicity of the synthetic route of a porous additive is considered in the literature as an important reason for developing new type of materials [24]. Therefore, highly energy and time efficient synthesis method of IR-PAN is an additional comparative property.

Comment 7. The particle size distribution of the ground IR-PAN-aM sample is monomodal with a fairly narrow particle size distribution of 500 to 800 nm obtained from the original IR-PAN-a containing three groups of particles: small particles, medium, and large (100, 800, and 8000 nm). Could you explain the phenomenon in grinding for the disappearance of the smallest particles, 100-500nm? In this procedure, it is ground and subsequently suspended in chloroform, what is the objective of this last step? These aspects could be included in the methodology and in the discussion.

Answer 7. In order to reduce the loading of the sample into a ball mill (400 ml), we used not a dry IR-PAN-a powder, but a suspension of IR-PAN-a in chloroform (0.1 wt %). After finishing the grinding process, the IR-PAN-aM suspension in chloroform was removed from the mill and separated from the ceramic grinding balls. The balls were rinsed with pure chloroform to remove IR-PAN-aM particles from their surface, and the resulting wash was mixed with the main suspension. The resulted suspension of IR-PAN-aM in chloroform was further used to prepare samples of TFC membranes CM-4, CM-5, and CM-6. It can be assumed that small particles of 100-500 nm, most likely, remained adsorbed on the surface of the mill.

Comment 8. The discussion of results on the impact of the size and %wt of the fillers on the thickness of the composite membrane should be better presented. Membranes with ground IR-PAN-aM decreases the thickness with respect to modification with IR-PAN-a, however, it is not understood how it can be thinner than the pristine sample (CM-7). Moreover, how does the incorporation of 20%wt reduce the thickness with respect to 10 and 30% wt? What are the arguments for these trends? are these differences considered significant? These tendencies must be correlated with the performance of the membranes.

Answer 8. The initial IR-PAN-a sample had a wide particle size distribution (from 90 to 6000 nm); the particles from the largest fraction had the size comparable to the thickness of the selective layer usually produced from a 0.5 wt % PTMSP solution in chloroform at a porous support speed of 0.3 m/min. Therefore, the thickness of the selective layer for the CM-1 - CM-3 samples was increased by reducing the speed of a porous support. For samples CM-4 - CM-6, a speed of 0.3 m/min was used, and the average thickness was from 0.8 to 1.0 μm, which is less than the distribution range of IR-PAN-aM particles (from 500 to 800 nm). The thickness of the CM-7 sample with pure PTMSP was 1.2 μm at the same support pulling speed as for the CM-4 - CM-6 samples. Such a discrepancy of several tenths of a micron is, most likely, an error in the deposition technique.

Comment 10. There is no clear trend or argument for the impact of size and percentage filler on gas permeance and selectivity. In this sense, conclusions should include these relevant arguments of the study.

Answer 10. The following changes have been made to the text of the revised manuscript:

Reviewing Table 4 one should notice that IR-PAN particle size and particle size distribution have a marked effect on the stability of the transport characteristics of TFC membranes. For IR-PAN-a sample with a wide particle size distribution and the presence of three particle sizes, including large ones (6000 nm) (Fig. 4), there is no clear pattern between the amount of the additive and the stability of the membrane transport characteristics. Comparison of membranes CM-1 (10 wt % IR-PAN-a) and CM-7 (without additive) indicates that the introduction of 10 wt % IR-PAN-a negatively affects the stability of the permeability of the composite membrane. Further introduction of 20 and 30 wt % IR-PAN-a leads first to an increase and then to a decrease in the stability of the permeability of TFC membrane (membranes CM-2 and CM-3, respectively). There is no logic in the relative ideal selectivity value, since this value should be higher for the CM-3 membrane, and not for the CM-2 membrane. On the other hand, the use of the IR-PAN-aM sample with a narrow particle size range of 500 - 800 nm (Fig. 4) leads to a steady improvement in the stability of membrane permeability in the entire studied range of additive concentrations (10, 20, and 30 wt %, membranes CM-4, CM-5 and CM-6, respectively). Moreover, the minimum studied additive content (10 wt %) ensured high gas transport characteristics and a minimal aging effect of TFC membranes. Further increase in the additive content leads to a slight increase in the aging effect (albeit insignificant), as evidenced by a decrease in the P′ from 30.2 to 27.2%.

Therefore, monomodal and narrow particle size distribution of highly porous activated IR-pyrolyzed PAN is a key factor for production of TFC membranes with reduced aging. In sum, 30 % of the initial permeance with almost unchanged membrane selectivity (α' = 1.1) after 11000 hours of aging at ambient laboratory conditions were preserved by introduction of the IR-PAN-aM particles with size of 500-800 nm into the selective layer from cross-linked PTMSP/PEI with thickness of 1000 nm.

Reviewer 2 Report

  • Experimental details are required even published before or mentioned with a reference.
  • “wt.” is better to be “wt”.
  • The purpose of the study is not clearly introduced and what is the purpose of aging?
  • Usually, the aging affects the crystal structure of materials, unfraternally there are no measurements related such as the XRD.
  • Mechanical measurements are required as well.

The present work could be important for readers of Membranes; however, I can’t recommend the publication of this work in its present form. There is much work that should be done to improve the manuscript, and major revisions are needed.

Author Response

Thank you very much for your time and effort to review our manuscript. We have made our best to improve our manuscript according to your comments.

  • Experimental details are required even published before or mentioned with a reference.

The experimental part of the article was extended.

  • “wt.” is better to be “wt”.

The correction has been done

  • The purpose of the study is not clearly introduced and what is the purpose of aging?

The goal of this work was specified at the end of the introduction:

«With this in mind, in this work IR-pyrolyzed PAN was investigated for the first time as a new type of filler to prevent physical aging of the PTMSP-based mixed matrix membranes.».

As a highly permeable glassy polymer with a high proportion of non-equilibrium free volume, PTMSP is characterized by the problem of physical aging, i.e. decrease in permeability over time [6. Tasaka, S., Inagaki, N., Igawa, M. Effect of annealing on structure and permeability of poly [(l-trimethylsilyl)-l-propyne]. J. Polym. Sci., Part B: Polym. Phys. 1991, 29(6), 691-694. https://doi.org/10.1002/polb.1991.090290607 and other reference 7-32]. Solving the problem of stabilization of its transport characteristics will allow this material, which is unique in its characteristics, to find practical application in membrane technology.

  • Usually, the aging affects the crystal structure of materials, unfraternally there are no measurements related such as the XRD.

This work was focused more on the aging of TFC membranes and the study of their properties. XRD investigations of composite membrane samples with a thin (~ 1 μm) selective layer are not possible due to the detection of a signal emanating from the support. Earlier we explored the change in the structure of PTMSP top-layer of 5 and 50 μm coated on the silicon support during the aging with wide-angle X-ray scattering (WAXS) [Bakhtin, D.S., Kulikov, L.A., Legkov, S.A., Khotimskiy, V.S., Levin, I.S., Borisov, I.L., Maksimov, A.L., Volkov, V.V., Karakhanov, E.A., Volkov, A.V. Aging of thin-film composite membranes based on PTMSP loaded with porous aromatic frameworks. J. Membr. Sci. 2018, 554, 211-220. https://doi.org/10.1016/j.memsci.2018.03.001.]. The change of the intensity of main peak upon annealing demonstrated that 50 μm PTMSP films were less influenced by the in-plane orientation of polymer near the substrate and in the skin-layer; whereas 5 μm PTMSP films, as well as all loaded samples (5 and 50 μm), were more affected by such anisotropy in macrochains packaging.

  • Mechanical measurements are required as well.

The study of mechanical properties of as-cast PTMSP-based samples (without and with the addition of PAF-11) was performed by dynamic mechanical analysis (DMA) in the stretch mode within the temperature range of 20–200 °C (heating rate is 2 °C/min). The results were published in our recent paper [Volkov, A.V., Bakhtin, D.S., Kulikov, L.A., Terenina, M.V., Golubev, G.S., Bondarenko, G.N., Legkov, S.A., Shandryuk, G.A., Volkov, V.V., Khotimskiy, V.S., Belogorlov, A.A., Maksimov, A.L., Karakhanov, E.A. Stabilization of gas transport properties of PTMSP with porous aromatic framework: Effect of annealing. J. Membr. Sci. 2016, 517, 80-90. https://doi.org/10.1016/j.memsci.2016.06.033.]. It was shown that there was a certain improvement of Young's modulus of mixed matrix materials with the addition of PAF-11. However, the difference in mechanical properties of dense films with varied PAF-11 loading became less pronounced at a higher temperature.

Reviewer 3 Report

The study highlighted the aging of TFC membrane that is based on cross-linked PTMSP/PEI loaded with highly porous carbon nanoparticles. It is indeed important to know the performance of the membrane as it aged. However, before publishing in Membranes, the following concerns should be addressed:

  1. In abstract, the author mentioned “lead to increase of permeance…”, please show what type of feed or gas.
  2. Please briefly discussed more the IR-pyrolyzed of PAN, not only mentioning that the protocol is similar to earlier work.
  3. Section 2.4.1, why the filler content is 10wt%?
  4. How the author assures that the large agglomerates is the particles? Is it not a dirt?
  5. Please describe why the morphology of Figure 2b is like that.
  6. The author used microporous support, is it possible that some PTMSP penetrate inside the support?
  7. Can the author provide a table of comparison for their membrane and other membrane in literature, base on performing the membrane regarding the aging time and testing condition?

Author Response

Thank you very much for your time and effort to review our manuscript. We have made our best to improve our manuscript according to your comments.

  1. In abstract, the author mentioned “lead to increase of permeance…”, please show what type of feed or gas.

Three gases N2, O2 and CO2 have been specified in the abstract.

2. Please briefly discussed more the IR-pyrolyzed of PAN, not only mentioning that the protocol is similar to earlier work.

The following protocol of IR-PAN synthesis has been added in the text:

In accordance with the protocol described in [37], PAN was first treated at 200 °C under IR heating in air for 20 min. The obtained powders were impregnated with KOH aqueous solution (ratio of pre-carbonized polymer:KOH=1:1 w/w). The suspension was exposed for 24 h and dried at 80 °C in vacuum until constant weight followed by IR heating stage at 800 °C for 2 min in a nitrogen atmosphere.

3. Section 2.4.1, why the filler content is 10wt%? 10%

The following explanation was added into in the section 3.1:

"Recently, we demonstrated that gas transport characteristics of the PTMSP dense membrane (films, thickness 30–40 µm) containing 10 wt % of PAF-11 became stable upon annealing at 100 °Ð¡ within the short time interval (100–200 h) [19]. However, for two other samples (1 and 5 wt %), gas permeability gradually decreased with time. In this study 10 wt % addition of IR-PAN-a was chosen to conduct a correct comparison of the effectiveness of a new additive."

It should be noted that the additive concentration of 10 wt % is the most frequently found in the investigation of aging of PTMSP/PAF mixed matrix membranes [Lau, C.H., Nguyen, P.T., Hill, M.R., Thornton, A.W., Konstas, K., Doherty, C.M., Mulder, R.J., Bourgeois, L., Liu, A.C.Y., Sprouster, D.J., Sullivan, J.P., Bastow, T.J., Hill, A.J., Gin, D.L., Noble, R.D. Ending Aging in Super Glassy Polymer Membranes. Angew. Chem. 2014, 53(21), 5322-5326. https://doi.org/10.1002/anie.201402234; Lau, C.H., Konstas, K., Doherty, C.M., Kanehashi, S., Ozcelik, B., Kentish, S.E., Hill, M.R. Tailoring Physical Aging in Super Glassy Polymers with Functionalized Porous Aromatic Frameworks for CO2 Capture. Chem. Mater. 2015, 27(13), 4756-4762. https://doi.org/10.1021/acs.chemmater.5b01537].

4. How the author assures that the large agglomerates is the particles? Is it not a dirt?

SEM images of pristine PTMSP dense membrane were added to fig.3c in revised version. Traces of fine dust can be seen on the surface of pristine PTMSP dense membrane. However, the amount and the size of inclusions on the image of the surface of mixed matrix membrane indicates that these are the agglomerates of 10 wt % IR-PAN-a. The following comment has been added to the text:

"Dense PTMSP membranes without the addition of IR-PAN-a were also made. SEM images of pristine PTMSP dense membrane are also presented in Figure 3. Traces of fine dust can be seen on the surface of pristine PTMSP dense membrane (Fig. 3c) and large agglomerates of particles up to several microns in size are visible on the surface and in the volume of the PTMSP membrane containing 10 wt % IR-PAN-a (Figs. 3 a, b)."

5. Please describe why the morphology of Figure 2b is like that.

The inhomogeneous structure in the lower part of the micrograph of the surface cleavage is a result of a sample preparation defect.

6. The author used microporous support, is it possible that some PTMSP penetrate inside the support?

Prior to coating, the support was impregnated with ethanol and then with water in order to prevent penetration of the polymeric layer into the pores of the MFFK-1 support.

7. Can the author provide a table of comparison for their membrane and other membrane in literature, base on performing the membrane regarding the aging time and testing condition?

To the best of our knowledge, most of the reported studies on the physical aging of PTMSP material were carried out with the dense films with the thickness from 30 to 150 μm. We summarized the published data from more than 30 publications in our recent article (see Table 1, ref. 49). For thick membranes, the permeability coefficient is a relevant property for comparison, since it characterize the material itself. There are a few publications on aging of TFC membranes. Besides, the TFC membrane performance is characterized by permeance. So it is difficult to make comparison of different membranes with different selective layer thickness.

Round 2

Reviewer 1 Report

The authors have effectively carried out all the suggested corrections and answered/clarified the consulted doubts.

Therefore, I suggest that this article be accepted for publication.

Reviewer 2 Report

The authors have addressed all my comments. Therefore I recommend the publication in Membrane as it is.